# Heavy Vacuum Gas Oil Upregulates the Rhamnosyltransferases and Quorum Sensing Cascades of Rhamnolipids Biosynthesis in *Pseudomonas* sp. AK6U

**DOI:** 10.3390/molecules26144122

**Published:** 2021-07-06

**Authors:** Sarah A. Alkhalaf, Ahmed R. Ramadan, Christian Obuekwe, Ashraf M. El Nayal, Nasser Abotalib, Wael Ismail

**Affiliations:** 1Environmental Biotechnology Program, Life Sciences Department, College of Graduate Studies, Arabian Gulf University, Manama 26671, Bahrain; soz-82@hotmail.com (S.A.A.); ashrafmt@agu.edu.bh (A.M.E.N.); nasseraa@agu.edu.bh (N.A.); 2Health Biotechnology Program, Life Sciences Department, College of Graduate Studies, Arabian Gulf University, Manama 26671, Bahrain; ahmedrr@agu.edu.bh; 3Department of Biological Sciences, Faculty of Science, Kuwait University, Kuwait 12037, Kuwait; okeyobuekwe@hotmail.com

**Keywords:** biosurfactants, congeners, surface tension, rhamnose, 3-(3-hydroxyalkanoyloxy) alkanoic acids

## Abstract

We followed a comparative approach to investigate how heavy vacuum gas oil (HVGO) affects the expression of genes involved in biosurfactants biosynthesis and the composition of the rhamnolipid congeners in *Pseudomonas* sp. AK6U. HVGO stimulated biosurfactants production as indicated by the lower surface tension (26 mN/m) and higher yield (7.8 g/L) compared to a glucose culture (49.7 mN/m, 0.305 g/L). Quantitative real-time PCR showed that the biosurfactants production genes *rhlA* and *rhlB* were strongly upregulated in the HVGO culture during the early and late exponential growth phases. To the contrary, the rhamnose biosynthesis genes *algC*, *rmlA* and *rmlC* were downregulated in the HVGO culture. Genes of the quorum sensing systems which regulate biosurfactants biosynthesis exhibited a hierarchical expression profile. The *lasI* gene was strongly upregulated (20-fold) in the HVGO culture during the early log phase, whereas both *rhlI* and *pqsE* were upregulated during the late log phase. Rhamnolipid congener analysis using high-performance liquid chromatography-mass spectrometry revealed a much higher proportion (up to 69%) of the high-molecularweight homologue Rha–Rha–C_10_–C_10_ in the HVGO culture. The results shed light on the temporal and carbon source-mediated shifts in rhamonlipids’ composition and regulation of biosynthesis which can be potentially exploited to produce different rhamnolipid formulations tailored for specific applications.

## 1. Introduction

Biosurfactants are green surface-active microbial products which are getting increasing interest due to their superior physicochemical properties and environmental compatibility as compared to synthetic (petroleum-based) surfactants [1,2]. In addition, biosurfactants have a broad range of environmental and industrial applications such as bioremediation, soil washing, biocontrol and application of fertilizers, enhanced oil recovery, cosmetics, pharmaceuticals, antimicrobial agents, foods and beverages, etc. [3,4,5,6,7]. Biosurfactants are structurally diverse and can be composed of proteins, peptides, fatty acids, phospholipids and carbohydrates. Glycolipids are the most common class of biosurfactants which include the well-known rhamnolipids produced by several *Pseudomonas* spp. [8,9].

Rhamnolipids are composed of one or two L-rhamnose units bound by a β-glycosidic linkage mostly to two fatty acid moieties linked through an ester bond; however, rhamnolipids containing only a single fatty acyl moiety were also reported [10]. The length of the fatty acid chains can vary from C_8_ up to C_14_, though the dominant component is C_10_–C_10_ [8,11]. Almost 25 rhamnolipid congeners have been described in *P. aeruginosa* with various chain lengths and/or extent of saturation [8,12] (and references therein). *P. aeruginosa* produces two major types of rhamnolipid biosurfactants: monorhamnolipids (l-rhamnosyl-β-hydroxydecanoyl-β-hydroxydecanoate, Rha–C_10_–C_10_) and dirhamnolipids (l-rhamnosyl-l-rhamnosyl-β-hydroxydecanoyl-β-hydroxydecanoate, Rha–Rha–C_10_–C_10_) (Appendix A) [13]. Rhamnolipids biosynthesis proceeds through three sequential reactions. The RhlA enzyme catalyzes the synthesis of the fatty acid dimer moiety of rhamnolipids as free 3-(3-hydroxyalkanoyloxy)alkanoic acids (HAAs), while the two rhamnosyl transferases, RhlB and RhlC, catalyze the transfer of dTDP-l-rhamnose to either HAAs, or a previously generated monorhamnolipid, respectively [8,12] (and references therein). The fatty acid components of rhamnolipids are recruited from the fatty acid biosynthesis and β-oxidation pathways, while the sugar moiety, dTDP-l-rhamnose, is produced from glucose with the AlgC and RmlBDAC enzymes [8,12]. Biosynthesis of rhamnolipids in *P. aeruginosa* is regulated by a complex and highly dynamic interplay between a multitude of genetic as well as environmental/nutritional factors including many quorum-sensing cascades such as the Rhl, Las and Pqs systems (Appendix A). In *P. aeruginosa*, the Las and Rhl quorum-sensing systems depend on acyl-homoserine lactone derivatives as signal molecules (autoinducers), such as *N*-3-oxododecanoyl-homoserine lactone (3-oxo-C_12_-HSL) and *N*-butanoyl-homoserine lactone (C_4_-HSL), respectively. These signal molecules are synthesized by the autoinducer synthases LasI and RhlI and bind to their cognate transcriptional regulators, LasR and RhlR, respectively, for regulation of the expression of several genes including the rhamnolipids biosynthesis genes [8,12].

Despite extensive efforts, biosurfactants have not been able to compete economically with their synthetic counterparts mainly due to high production costs which can be up to 12 times higher than that of the conventional petroleum-based surfactants [14]. To overcome this barrier, low-value carbon-rich feed stocks such as agricultural and industrial wastes/residues have been used by many investigators as carbon sources for biosurfactants production, thus replacing synthetic culture medium which constitutes ~30% of the production costs [9,14,15]. Moreover, hydrophobic inducers such as vegetable oils and different fatty acids were used to boost the productivity of biosurfactants and their effect exceeded that of hydrophilic compounds such as sugars [9,14]. However, it is not sufficiently understood how these different carbon sources affect the expression of the rhamnolipids biosynthetic genes and the underlying regulatory networks. Furthermore, knowledge on rhamnose biosynthesis from non-carbohydrate carbon sources, such as oils and hydrocarbons, is largely lacking. Undoubtedly, better understanding of these processes will enable higher yields and productivities of biosurfactants and, consequently, improve the economic viability.

Although petrochemical wastes and processing residues such as refinery wastes, sludge and difficult-to-refine hydrocarbon streams are potential substrates for biosurfactants production, they have been rarely investigated [16,17,18]. Recently, we have shown that a *Pseudomonas* sp. AK6U strain can utilize heavy vacuum gas oil (HVGO) as the sole carbon and sulfur source for a high-yield production of rhamnolipids concomitant with biocatalytic upgrading of the heavy oil substrate. Moreover, chemical analysis of the produced rhamnolipids congeners revealed, unexpectedly, the dominance of the rarely occurring dirhamnolipid Rha–Rha–C_12_–C_12_ [19]. These results raised the questions of whether HVGO affected the expression of the *rhlABC* genes and/or those of the involved quorum sensing networks, and how rhamnose is produced in the HVGO culture which lacks exogenous sugars. In addition, it is not known if and how the rhamnolipids congener profile of the HVGO culture is different from that produced using the conventional carbon source glucose, and how the incubation time might affect the composition of the produced rhamnolipids. 

To address these questions and fill the underlying knowledge gaps, we performed comparative real-time quantitative PCR (RT-qPCR) on AK6U cultures grown on either HVGO or glucose as the sole carbon source at different growth phases. We also fingerprinted the rhamnolipid congener’s profile in both cultures using high-performance liquid chromatography-mass spectrometry (HPLC-MS). The results uncover the effect of HVGO on the expression of the rhamnolipid (*rhlABC*) and rhamnose (*algC*, *rmlAC*) biosynthetic genes in addition to those of the quorum sensing systems (*lasI*, *rhlI*, *pqsE*) and highlight major deviations in the rhamnolipid congeners’ profile triggered by HVGO.

## 2. Results

### 2.1. The AK6U Strain Utilizes HVGO as a Sole Carbon Source for Growth and Biosurfactants Production

As shown in Figure 1, the turbidity of both the glucose and HVGO cultures increased with time and there was a temporal increase in foaming in both cultures. However, with a specific growth rate (*µ*) of 0.33 ± 0.04 h^−1^, the glucose culture was ~5 times faster than the HVGO culture (*µ* = 0.07± 0.01 h^−1^). Moreover, in the HVGO culture the consistency of the oil changed with time and it was eventually emulsified in the culture medium. Altogether, these changes in culture turbidity, foaming and oil emulsification indicate that the AK6U strain grew on and utilized HVGO for production of surface-active compounds. This was confirmed by the biomass and surface tension measurements as shown in Figure 2.

### 2.2. HVGO Stimulates Biosurfactants Production

The surface tension of cell-free and oil-free culture supernatants decreased with time in both the glucose and HVGO cultures. However, the overall profile of the surface tension was different between both cultures (Figure 2). First, the decrease in surface tension in the HVGO culture was stronger than in the glucose culture. The HVGO culture reached a minimum surface tension of 26 mN/m, whereas the glucose culture recorded a minimum of 49.7 mN/m. Second, the decrease in surface tension occurred faster in the glucose culture (reached a minimum within 12 h) than in the HVGO culture (reached a minimum within 120 h). Nonetheless, in both cultures no further reduction in surface tension could be detected beyond the time point where the minimum was attained. As shown in Figure 2C, the crude biosurfactants yield from the HVGO culture was ~7-fold and 25-fold higher than that obtained from the glucose culture at the late log and stationary phases, respectively. In addition, biosurfactants yield from the stationary phase HVGO culture was ~7-fold higher than that of the late lag phase culture. In contrast, the glucose culture’s biosurfactants yield during the stationary phase was only 2-fold higher than that obtained from the late log phase culture.

### 2.3. The Rhamnolipids Composition in the HVGO Culture Is Different from That of the Glucose Culture

The occurrence and relative abundance of the HAAs of rhamnolipids detected in cultures of strain AK6U are shown in Table 1. HAAs diversity and relative abundance differed depending on the carbon source and incubation time. No HAAs were detected in the late log phase cultures. In the stationary phase culture of glucose, only four different HAAs were detected, the majority of which (three) are composed of short-chain acyl group derivatives of octanoic acid which constituted more than 99% (relative abundance) of HAAs present in that culture. The dioctanoic acid moiety (–C_8_–C_8_) formed the most abundant (76.01%) fraction of HAAs. Only a minor amount (0.35%) of dodecanoyl-β-oxydodecenoic acid moiety (–C_12_–C_12:1_) was detected in the glucose culture. Unlike the glucose culture, a higher number (14) of HAAs was present in the stationary phase culture of HVGO. In addition, the HVGO culture produced several abundant (>7%) HAAs moieties containing long-chain acyl groups. However, as observed in the glucose culture, the most abundant (42.98%) HAAs in the HVGO culture was the short-chain acyl group derivatives (–C_8_–C_8_).

In analogy with the HAAs profile, the level, diversity and relative abundance of the rhamnolipid congeners differed depending on the carbon source and growth phase (Table 2). On glucose, 15 different rhamnolipid congeners were detected in the stationary phase culture. The lower-molecularweight homologues: monorhamnosyl monooctanoate and dirhamnosyl dioctanoate derivatives were the most dominant. In the glucose culture, the most abundant (86.4%) rhamnolipid congener was the dirhamnolipid Rha–Rha–C_8_–C_8_. To the contrary, the high-molecularweight congeners containing dirhamnose and long-chain dialkanoates were produced in very low numbers and abundance.

Unlike the HAAs profile, at the late log phase of the HVGO culture a total of 17 rhamnolipid congeners were produced in detectable amounts. However, the detectable number of the congeners in the same culture declined to 16 during the stationary phase. Although the numbers of rhamnolipid congeners found at the late log (17) and stationary phase (16) were comparable, their profiles were different. Four congeners: Rha–C_8_, Rha–Rha–C_8_, Rha–Rha–C_8_–C_8_ and Rha–Rha–C_12:1_–C_12_ were only found in the stationary phase culture, while the five congeners: Rha–C_10_, Rha–Rha–C_12:1_, Rha–C_12:1_–C_10_, Rha–C_12_–C_10_ and Rha–C_10_–C_12_ were detected in the late log phase culture. Irrespective of the incubation time, and unlike the glucose culture, the rhamnolipid congeners produced from HVGO were dominated (52.94 and 68.75% in the late log and stationary phase, respectively) by the higher-molecularweight species (the dirhamnosyl homologues). In relative amounts, the most abundant congener was dirhamnosyl-β-hydroxydecanoyl-β-hydroxydecanoic acid (Rha–Rha–C_10_–C_10_), which accounted for 50.0 and 69.3% of all congeners produced at the late log and stationary phase, respectively. The monorhamnosyl homologue (Rha–C_10_–C–_10_) was the most abundant congener produced at the late log phase (14.33%) and declined to 3.43% in the stationary phase culture. With the exception of Rha–Rha–C_10_–C_10_, the relative abundance of all dirhamnolipid congeners produced from HVGO declined when the culture moved from the late log to the stationary phase of growth.

### 2.4. Isolation of RNA and cDNA Synthesis

RNA was isolated from biomass collected at the early and late log phases of both the glucose and HVGO cultures and its integrity was verified by gel electrophoresis (Appendix A). To check for the presence of genomic DNA contamination in the RNA preparations, samples of RNA were used as a template in PCR to amplify fragments of the *rhlA* gene with gene-specific primers. All PCR assays failed to amplify any products, while a PCR with genomic DNA from the AK6U strain produced the expected *rhlA* gene amplicon (Appendix A), thus confirming that all RNA preparations were free of genomic DNA. The cDNA was then synthesized from the mRNA transcripts and checked for validity as a template in PCR assays to amplify a fragment from the *rhlA* gene. As shown in Appendix A, the target fragment was successfully amplified from all cDNA preparations. 

The 16S rRNA gene was used as a reference to quantify the relative expression of nine genes involved in rhamnolipid biosurfactants production (*rhlA*, *rhlB*, *rhlC*, *algC*, *rmlA*, *rmlC*, *pqsE*, *rhlI* and *lasI*). The reaction efficiency of the RT-qPCR assays was in the range 99–100%. Moreover, melt curve analysis revealed the absence of unspecific PCR products for all the analyzed genes in both the glucose and HVGO cultures (Appendix A).

### 2.5. HVGO Induces the Expression of the Rhamnolipids Biosynthesis Genes

To unravel whether the increased rhamnolipids production in the HVGO culture was due to upregulation of the key biosynthetic genes, and to examine the effect of the incubation time on gene expression, we performed comparative qPCR on the *rhlABC* genes using mRNA transcripts from both the glucose and HVGO cultures at different growth phases (Figure 3). Interestingly, as compared to the glucose culture, the HVGO culture exhibited significantly higher relative expression of *rhlA* at both the early (28-fold) and late log (33-fold) phases of growth (*p* ˂ 0.0001). There was no significant difference in *rhlA* expression (*p* = 0.99) between the early and late log cultures of glucose. In contrast, in the HVGO culture, the *rhlA* gene was significantly downregulated during the late log phase as compared to the early log phase culture (*p* = 0.0005).

The *rhlB* gene exhibited an expression profile similar to that of *rhlA* (Figure 3). For both the early and late log phases, the *rhlB* gene expression was significantly higher in the HVGO culture (19- to 41- fold, *p* ˂ 0.005). However, there was no significant difference in *rhlB* expression between the early log and late log glucose cultures, and the early log and late log HVGO cultures.

The *rhlC* expression profile was different from that of the *rhlAB* genes. During the early log phase, there was no significant difference (*p* > 0.05) between the glucose and HVGO cultures (Figure 3). In contrast, during the late log phase, *rhlC* was significantly upregulated (3.5-fold) in the HVGO culture (*p* = 0.0001). Moreover, the *rhlC* expression level in the late log HVGO culture was significantly higher (~5-fold, *p* = 0.0001) than that of the corresponding early log phase culture.

### 2.6. Downregulation of the Rhamnose Biosynthesis Genes in the HVGO Culture

Since rhamnolipids production in the HVGO culture was much higher than that observed in the glucose culture, we expected to see increased expression of the rhamnose biosynthesis genes in line with the increased expression of the *rhlABC* genes. Therefore, it was surprising to see that the rhamnose biosynthetic genes (*rmlA*, *rmlC*, *algC*) were downregulated in the HVGO culture as compared to the glucose ones. As shown in Figure 4, the *algC* expression level in the glucose culture was significantly higher (*p* ˂ 0.015) than that of the HVGO culture during both the early (6.5-fold) and late log (5.5-fold) growth phases. However, there was no significant difference among the HVGO cultures and the glucose cultures from both growth phases (*p* > 0.05). The expression profile of the *rmlA* gene was similar to that of the *algC*. For both growth phases, the HVGO culture had a significantly lower *rmlA* expression (9–14-fold) as compared to the glucose culture (*p* ˂ 0.0002). Moreover, in the late log glucose culture, *rmlA* was downregulated (*p* = 0.004) as compared to the corresponding early log culture. In case of HVGO, there was no significant difference (*p* = 0.9) in the *rmlA* expression level between the early and late log cultures. During the early log phase, the glucose culture had significantly higher level of *rmlC* than the HVGO culture (23-fold, *p* ˂ 0.0001) (Figure 4). However, there was no significant difference in *rmlC* expression between the glucose and HVGO cultures of the late log phase (*p* = 0.14). Nonetheless, during the late log phase the transcription of *rmlC* was significantly higher in the HVGO culture (12-fold) compared to the early log phase culture (*p* = 0.0017). The *rmlC* expression in the glucose culture of both the early and late log phases was not significantly different (*p* > 0.05).

### 2.7. HVGO Remodels the Expression of the Quorum Sensing Genes

The quorum sensing genes which regulate rhamnolipids biosynthesis also exhibited an interesting expression pattern (Figure 5). Although there was no significant difference in the level of *rhlI* transcripts between the glucose and HVGO cultures during the early log phase (*p* = 0.38), the *rhlI* gene was significantly upregulated (~6-fold) in the late log HVGO culture (*p* ˂ 0.0001). The expression of *rhlI* did not vary significantly with the incubation time in the glucose culture (*p* = 0.96). The expression pattern of *pqsE* was very much similar to that of *rhlI* (Figure 5). The late log HVGO culture had the highest expression level (*p* ˂ 0.0001) that was 34-fold higher than that of the glucose cultures of both growth phases.

Moreover, the late log HVGO culture had a *pqsE* expression level that was 95-fold higher than that of the corresponding early log phase culture. The glucose cultures from both growth phases exhibited similar expression levels (*p* = 1). In contrast to *rhlI* and *pqsE*, the highest expression of the *lasI* gene occurred during the early log phase of the HVGO culture. In this culture, the *lasI* expression level was significantly higher (20-fold) than that of the early log glucose culture (*p* ˂ 0.0001). During the late log phase, the *lasI* gene was significantly downregulated (*p* ˂ 0.0001) in the HVGO culture. The glucose cultures of both the early and late log growth phases had similar expression level of *lasI* (*p* = 1) which was significantly lower than that of the early log HVGO culture.

## 3. Discussion

In our previous study [19] we showed that *Pseudomonas* sp. AK6U can utilize HVGO as the sole carbon and sulfur source to produce rhamnolipid biosurfactants. Nonetheless, it was not clear whether HVGO can stimulate biosurfactants production compared to hydrophilic substrates such as glucose. In the current study, the stronger reduction in surface tension together with the higher yield of biosurfactants in the HVGO culture showed that HVGO was a better substrate for biosurfactants production than glucose [8,19,20,21,22]. Rhamnolipids are known to reduce the surface tension to values between 28 and 30 mN/m [7]. The minimum surface tension value measured in the HVGO culture (26 mN/m) is relatively lower than reported values for the AK6U strain on either HVGO (30.6 mN/m) or organosulfur compounds (30–33 mN/m) [19,21,22]. Bacteria usually produce biosurfactants to make hydrophobic substrates, such as oils and hydrocarbons, more accessible for biodegradation and growth [20,23].

The strong upregulation of the *rhlABC* genes in the HVGO culture explains the observed higher biosurfactants yield compared to the glucose culture. Ismail et al. [21] reported that cultures of the strain AK6U grown on the organosulfur compound dibenzothiophene as a sole sulfur source had significantly higher expression levels of the *rhlABC* genes as compared to cultures grown with inorganic sulfate. The higher level of the *rhlAB* genes during the early log phase is in line with the assumption that biosurfactants production in the HVGO culture started at the beginning of the lifespan, following a growth-associated production mode [21,22,24,25]. This is also coherent with the observed reduction in surface tension. Since *rhlC* transcription increased in the HVGO culture toward the stationary phase, it might explain the observed stimulation of rhamnolipids production during the stationary phase of the HVGO culture. It also conforms to the frequently reported observation that rhamnolipids production is triggered at the onset of the late log-early stationary phases of growth [26,27,28].

The notion that the carbon source is a key determinant of the biosurfactants’ congener profile [29,30] raised the question whether HVGO-derived rhamnolipids exhibited a congener composition different from that attained on glucose. The answer was revealed by the congener profile analysis showing distinct rhamnolipid mixtures in the HVGO and glucose cultures. Accordingly, it can be proposed that the differences in the expression of the *rhlABC* genes between the glucose and HVGO cultures reflect the observed variations in the rhamnolipids congener composition. As shown in this study, differences in congener composition can be not only in terms of the number of the rhamnose moieties, but also the chain length of the fatty acid components. For instance, Mata-Sandoval et al. [31] reported the production of rhamnolipids by *P. aeruginosa* UG2 using corn oil as a carbon source with the dirhamnolipid Rha–Rha–C_10_–C_10_ as the major congener (60.6 mol%). Other authors reported the monorhamnolipid Rha–C_10_–C_10_ as the dominant species in rhamnolipid mixtures produced by other strains [32,33]. We showed in our recent study that the rarely occurring dirhamnolipid congener Rha–Rha–C_12_–C_12_ dominated rhamolipids produced by the AK6U strain using HVGO as a carbon and sulfur source. Although the HVGO-derived rhamnolipids were dominated by the dirhamnose homologues, in agreement with our previous study [19], the most dominant congener (Rha–Rha–C_10_–C_10_) was different from that reported by Ismail et al. [19]. This could be due to the different quantity of HVGO used in both studies (10% in the current study versus 20% in the previous study). Moreover, Ismail et al. [19] added HVGO to the culture medium as both carbon and sulfur source, while HVGO was used only as a carbon source in the current study and sulfur was provided as inorganic sulfate.

The higher number of HAAs in the HVGO culture compared to the glucose culture could be attributed to the higher expression level of *rhlA* encoding the fatty acid dimers-synthesizing enzyme. The upregulation of *rhlC* in the HVGO culture during the late log phase suggests increased synthesis of dirhamnolipid congeners which might lead to uptake and exhaustion of HAAs and explains why HAAs were not detectable in the late log culture of HVGO. Uptake of extracellular HAAs and monorhamnolipid congeners from culture media for the production of dirhamnolipids containing two acyl groups was reported [34].

The decrease in the relative abundance of dirhamnolipids (except Rha–Rha–C_10_–C_10_) in the HVGO culture toward the stationary phase seems contradictory to the upregulation of *rhlC*. While the higher expression of *rhIC* in the late log phase of the HVGO culture may serve as a prelude for increased synthesis and larger number and diversity of dirhamnolipid congeners produced in the stationary phase, it does not explain the decline in the abundance of each dirhamnosyl dicarboxylic acid derivative when the growth was shifted to the stationary phase. A potential scenario is that dirhamnolipids may be degraded as the culture gets older. Recently, Wittgens et al. [34] studied rhamnolipid production in the heterologous host *P. putida* KT2440 and the native rhamnolipid producer *P. aeruginosa* PAI. The authors provided evidence thought to support hydrolytic cleavage of ester bonds between the two alkanoic acid groups contained in the final products of rhamnolipid biosynthesis as the source of rhamnolipids that contain one fatty acid moiety (monorhamno-monolipid and dirhamno-monolipid). Such postulated hydrolytic activity may explain the decline in the abundance of dirhamnolipids containing two acyl groups as seen in the stationary phase culture of HVGO with the formation of the mono-acyl homologues. The fatty acid resulting from the hydrolysis would presumably be recycled for enhanced synthesis of HAAs and, ultimately, rhamnolipids supported by the observed upregulation of *rhIABC* genes in the HVGO culture. In favor of the proposed rhamnolipid degradation, the dirhamnosyl-monodecanoate (Rha–Rha–C_10_), a possible hydrolytic product of Rha–Rha–C_10_–C_10_ which was detected in both the late log and stationary phases, was more abundant in the stationary phase than in the late log phase culture. This observation appears to support the assumption that decline in abundance of dirhamnosyl-dialkanoates derive from hydrolysis of ester bonds between the two acyl groups, as suggested by Wittgens et al. [34].

The higher expression of the rhamnose biosynthesis genes *algC*, *rmlA* and *rmlC* in the glucose culture agrees with their known role in rhamnose biosynthesis using glucose as a precursor. Aguirre-Ramirez et al. [35] studied the regulation of the *rmlBDAC* operon in *P. aeruginosa* PAO1 and showed that the expression of these genes was high throughout the culture’s lifetime and was further induced at the start of the stationary phase which coincided with the onset of rhamnolipids production. In contrast, the downregulation of the *algC*, *rmlA* and *rmlC* genes in the HVGO culture appears contradictory to the observed higher rhamnolipids yield which obviously requires higher content of rhamnose than the glucose culture, particularly in view of the increased expression of *rhlC* and biosynthesis of dirhamnolipids. These results raise the question: how rhamnose was produced in the HVGO culture that lacks glucose?

Although the data presented here do not provide a clear answer to this question, one can propose different possibilities. It can be assumed that in the HVGO culture, the strain AK6U depended partially on the *rml* pathway to produce rhamnose, and the bulk of rhamnose was synthesized via an alternative, yet unknown, pathway. This assumption is supported by the observed upregulation of the *rmlC* gene in the late log HVGO culture. However, if the *rml* pathway was functional in the HVGO culture, where does glucose come from to be converted to rhamnose? Glucose, or probably rhamnose, could be retrieved from cell surface polysaccharides which might be partially degraded as a stress response to the presence of HVGO as the sole carbon and energy source [36]. Alternatively, glucose might be released from the degradation of stored carbohydrates such as glycogen which was shown to be produced by some *Pseudomonas* spp. [37,38,39,40]. We may also hypothesize that glucose could be synthesized *de novo* via gluconeogenesis [41]. The catabolism of the hydrocarbon components of HVGO produces the central metabolites acetyl-CoA, pyruvate and succinate which are further metabolized in central metabolic pathways such as the tricarboxylic acid and glyoxylate cycles to produce the anabolic building blocks oxaloacetate and phosphoenolpyruvate. The latter is the starting building block of glucose via reversal of glycolysis [39,40]. This is one of the interesting aspects which deserves further in-depth investigation. However, it is a challenging task, knowing that sugar metabolism in pseudomonads operates via a complex network and dynamic interplay between the Embden-Meyerhof-Parnas, Entner-Doudoroff and pentose phosphate pathways, constituting the so-called EDEMP cycle [42].

The temporal changes in transcript levels of the quorum sensing genes are in line with previous studies and provide evidence for the involvement of the three quorum sensing systems *las*, *pqs* and *rhl* in the regulation of rhamnolipids production by the strain AK6U [12,43]. As such, the upregulation of the quorum sensing genes explains the upshift in the expression of the *rhlABC* genes which caused production of higher amount of biosurfactants in the HVGO culture. Many studies reported increased rhamnolipids production as a result of the upregulation of the *rhlABC* genes which was due to positive regulation by the quorum sensing systems. For instance, Bazire et al. [44] reported increase in *rhlA*, *rhlB* and *rhlC* transcripts concomitant with rhamnolipids production by *P. aeruginosa* during the stationary phase after the maximum level of the autoinducer C_4_-HSL (product of RhlI) was attained. Moreover, increased production of rhamnolipids by *P. aeruginosa* at the early stationary phase was attributed to RhlI-dependent upregulation of the *rhlABC* genes [45,46].

The three quorum sensing systems, *las*, *rhl* and *pqs*, constitute a highly complex regulatory network which operates in a finely tuned and intertwined hierarchy or cascade. The *rhl* system directly activates the expression of the *rhlABC* genes (rhamnolipids biosynthesis) and the *rmlBDAC* operon (L-rhamnose biosynthesis). The *pqs* system positively regulates the *rhl* system which represses the expression of the *pqs* genes. At the top of the hierarchy dominates the *las* system which activates both the *rhl* and *pqs* systems [12,35,43]. The expression profile of the *lasI*, *pqsE* and *rhlI* genes is consistent with this hierarchy. The *lasI* gene was strongly upregulated in the early log HVGO culture and its expression declined during the late log phase. In contrast, both the *pqsE* and *rhlI* genes followed the opposite expression profile, being highly expressed during the late exponential phase. This expression pattern indicates that the *las* system was activated first, which is consistent with its position at the top of the cascade, and subsequently itactivated the *pqs* and *rhl* systems. The *rhl* system eventually induced the expression of the *rhlABC* genes. Bazire et al. [44] studied the time course of production of the quorum sensing signal molecules produced by the three systems (*las*, *rhl*, *pqs*) in *P. aeruginosa*. The authors reported that the signal molecule 3-oxo-C_12_-HSL (product of LasI) reached its maximum concentration during the exponential growth phase, and was almost absent towards the onset of the stationary phase.

The LasR/3-oxo-C_12_-HSL signaling pathway was also reported to directly activate the expression of *rhlI*, and the *rhl* system is *las*-dependent [8,12,47]. This does not seem to be the case here because *rhlI* was not highly expressed in the early log HVGO culture, although *lasI* was strongly upregulated. The upregulation of *rhlA* and *rhlB* and biosurfactants production during the early log phase suggest that regulatory factors other than the *las* system were potentially controlling rhamnolipids production during the early log phase [12]. Dekimpe and Deziel [48] showed that the expression of the *rhl* system was maintained in a *lasR* mutant. Furthermore, *rhlR* is known to have four different transcription start sites [49]. In rich medium, the expression of *rhlR* is dependent on the LasR/3-oxo-C_12_-HSL signaling pathway. However, under nutrients shortage (e.g., phosphate starvation), the expression of *rhlR* is regulated from multiple promoters through different transcriptional activators such as Vfr and sigma factor σ^54^.

Though direct evidence is lacking, it is tempting to speculate that HVGO interfered with the dynamics of the quorum sensing systems, and, consequently, the expression of related genes. In the biphasic HVGO culture medium, the highly hydrophobic HVGO may act as a solvent or sink for the quorum sensing signals that alters their partition, and hence their concentration, in the aqueous phase, compared to the monophasic glucose culture. This scenario might change with time due to the strong emulsification of HVGO with the produced rhamnolipids. In addition, the toxicity of some HVGO components may potentially trigger different stress responses accompanied by alteration of the normal physiology and metabolism of the AK6U strain, including the transcription and translation machinery, thus adding a further level of complexity to the study of biosurfactants production under these conditions.

## 4. Materials and Methods

### 4.1. Bacteria

*Pseudomonas* sp.AK6U was used as a model bacterium. This strain was isolated and characterized in previous investigations at the laboratories of the Environmental Biotechnology Program-Arabian Gulf University. It produces rhamnolipid biosurfactants using glucose or HVGO as a carbon source and stimulates biosurfactants production when the conventional sulfur source, inorganic sulfate, is replaced by thiophenic sulfur compounds such as dibenzothiophene [19,21,22].

### 4.2. Culture Medium and Growth Conditions

Chemically defined medium (CDM) was prepared from stock solutions as shown in Appendix A. The carbon source was either glucose (10 mM) or HVGO (10%, vol/vol). Liquid cultures were routinely grown in 250 mL Erlenmeyer flasks containing 100 mL of the culture medium and incubated in an orbital shaker (180 rpm) at 30 °C. Uninoculated medium was routinely included as an abiotic control.

### 4.3. Biosurfactants Production by the AK6U Strain

To prepare a starter culture, a single colony of AK6U was picked from a fresh LB (Luria-Bertani)-agar plate (incubated for 48 h at 30°C) and inoculated into 100 mL of LB followed by overnight incubation. The biomass was harvested from 50 mL of the culture by centrifugation at 10,000 rpm for 5 min. The cell pellet was washed once in K-phosphate buffer (0.1 M, pH 7), and the washed cell pellet was resuspended in 10 mL of the same buffer. This cell suspension was used to inoculate the glucose and HVGO cultures as described below.

CDM was prepared in two sets (in biological triplicates) of one-litter baffled Erlenmeyer flasks each containing a final volume of 400 mL of the culture medium. One set of flasks contained glucose and the other contained HVGO as the sole carbon source. Each flask was inoculated with 2 mL of the cell suspension to give a biomass concentration of 0.5 g dry cell weight/L. All cultures including the uninoculated controls were incubated in an orbital shaker and culture samples were retrieved at time intervals for biomass measurement, monitoring biosurfactants production and isolation of RNA as described later. Growth was monitored by measuring the optical density at 600 nm (OD_600_) for the glucose cultures, while growth in the HVGO cultures was monitored by measuring the biomass dry weight. The cells were harvested from 25 mL of the cultures and dried at 105 °C for 20 h. Biosurfactants production was visually assessed by observing culture foaming and measuring the surface tension in cell-free culture supernatants.

### 4.4. Surface Tension Measurement

The surface tension of cell-free culture supernatants was measured at room temperature with a Kruss K100MK3 Tensiometer (Kruss, Germany) equipped with a platinum plate via the Wilhelmy plate method [21]. The tensiometer was calibrated by adjusting the measurement so that the surface tension of water is 72 mN/m.

### 4.5. Quantification of the Biosurfactants and Congener Analysis

Crude rhamnolipid biosurfactants were extracted from cell-free supernatants of the AK6U cultures grown either on glucose or HVGO. Culture samples were collected at the late log (for glucose after 12 h, for HVGO after 178 h) and stationary phase of growth (for glucose after 17.5 h, for HVGO after 329 h). Culture samples (50 mL) were centrifuged to collect cell-free and oil-free supernatants and the biosurfactants were extracted as described [19]. After solvent evaporation, the weight of the remaining residue was measured and used to calculate the biosurfactants yield (g/L). Analysis of the of HAAs and conger profile of rhamnolipids extracted from the glucose and HVGO cultures of AK6U at different time points was performed with HPLC-MS as described [22].

### 4.6. Isolation of RNA from the AK6U Cultures

Samples from the glucose and HVGO cultures were retrieved at different time intervals to harvest cells for total RNA isolation. For the glucose cultures, samples were collected after 7 h (early log phase) and 12 h (late log phase). For the HVGO cultures, samples were collected after 154 h (early log phase) and 178 h (late log phase). In total, six samples were collected for each time point (two samples from each biological replicate culture). All culture samples (1 mL each) were centrifuged in sterile Eppendorf tubes (10 min, 10,000 rpm, 4 °C) to harvest the cells. The supernatants were decanted and each cell pellet (~3 mg) was resuspended in 200 µL of lysozyme-TE buffer. Total RNA was isolated using RNeasy Mini kit (Qiagen, Germany) according to the manufacturer’s instructions. The isolated RNA was treated twice with DNase I to remove genomic DNA contaminants. To check for genomic DNA contamination, PCR was conducted with gene-specific primers (*rhlA*, Table 3) using the isolated RNA as a template and genomic DNA from the AK6U strain as a positive control. The PCR mixtures (20 µL) contained 1 µL of RNA or DNA, 1 µL of each primer (10 pmol/µL) and 10 µL PCR master mix (Qiagen, Germany). The PCR conditions were: 5 min for initial denaturation at 95 °C, followed by 35 cycles at 95 °C for 1 min, 59 °C for 30 s, 72 °C for 45 s and final extension 5 min at 72 °C.

### 4.7. Synthesis of Complementary DNA

Complementary DNA (cDNA) was synthesized from a normalized RNA quantity (1000 ng/µL) with High Capacity cDNA Reverse Transcription kit (ABI, Vernon, CA, USA) according to the manufacturer’s instructions. The RNA was reverse transcribed into cDNA in a total volume of 40 µL and the conditions for the reverse transcription reaction were: 25 °C for 10 min, 37 °C for 2 h and 85 °C for 5 min to inactivate the reverse transcriptase. The concentration and purity of cDNA were estimated using Biophotometer Plus (Eppendorf, Hamburg, Germany).

The validity of cDNA as a template for the qPCR assays was checked by conventional PCR assays using primers to amplify a fragment of the *rhlA* gene (Table 3). Genomic DNA from the strain AK6U was also tested in a positive control assay. The PCR assay (20 µL) contained 10 µL PCR master mix, 1 µL of each primer (10 pmol/µL) and 1 µL of cDNA or genomic DNA. The PCR conditions were: 5 min for initial denaturation at 95 °C, followed by 35 cycles at 95 °C for 1 min, 59 °C for 30 s, 72 °C for 45 s and final extension for 5 min at 72 °C.

### 4.8. Quantification of Gene Expression by RT-qPCR

Expression of the genes involved in the biosynthesis of rhamnolipids and its regulation in the AK6U strain was evaluated by quantifying mRNA transcripts with RT-qPCR using the 16S rRNA gene as a reference. The target genes included in this study were *rhlA*, *rhlB*, *rhlC* (biosynthesis of mono- and dirhamnolipids), *rmlA*, *rmlC*, *algC* (rhamnose biosynthesis), *lasI*, *rhlI* and *pqsE* (quorum sensing-mediated regulation of rhamnolipid biosynthesis) (Table 3). Primers specific for the *rhlAC*, quorum sensing and rhamnose biosynthesis genes were derived from the corresponding sequences in the genome of the *Pseudomonas aeruginosa* PAO1 strain (GenBank accession number NC_002516) with Primer 3 [51]. Primers of the 16S rRNA gene are based on the partial sequence of the 16S rRNA gene of the AK6U strain (GenBank accession number AB922602) [21]. For each time point, six cDNA preparations (obtained from three biological replicates) were tested in RT-qPCR using Rotor-Gene Q (Qiagen, Hilden, Germany). Each RT-qPCR assay (15 µL in nuclease-free water) contained 7.5 µL Rotor-Gene master mix (SYBR-Green PCR kit, Qiagen, Hilden, Germany), 0.5 µM of each primer and 500 ng of cDNA. The qPCR conditions were: 5 min for initial denaturation at 95 °C, followed by 40 cycles at 95 °C for 30 s, 58 °C (59 °C for the *rhlABC* genes) for 20 s and 60 °C for 45 s.

At the end of the PCR program, a melt curve analysis was run to check for the presence of unspecific PCR products by increasing the temperature gradually from 70 to 99 °C at a rate of 1 degree/step. The reaction efficiencies of each amplicon were calculated from the slope of calibration curves generated from 6-fold dilutions of genomic DNA. All qPCR assays were checked by agarose gel electrophoresis.

The 2^−ΔΔCT^ method for relative quantification of gene expression was applied to investigate the changes in target gene expression in AK6U cultures grown with either glucose or HVGO as a carbon source [52]. The C_T_ of each target gene was normalized to the C_T_ of the reference gene (16S rRNA) for both the glucose (calibrator) and the HVGO (test) cultures. The ΔC_T_ of the target gene in the HVGO culture was then normalized to that of the glucose culture. Eventually, the expression ratio was calculated (2^-ΔΔCT^ = normalized expression ratio).

### 4.9. Statistical Analysis

All results of growth and gene expression represent the average of at least triplicate experiments. Significance of the differences was tested via one-way analysis of variance (ANOVA) using the Tukey test (*p* = 0.05) with the JMP statistical software (version 10.0.2, SAS Corporation, Chicago, IL, USA).

## 5. Conclusions

*Pseudomonas* sp. AK6U responded to HVGO by reprograming the expression of the biosynthetic and regulatory machinery of rhamnolipids production. These temporal shifts in gene expression were apparently necessary to boost biosurfactants production and remodel the rhamnolipid composition as enabling tools for the biodegradation and utilization of the challenging sole carbon source HVGO. Although the results of this study can potentially be exploited to produce higher yields of biosurfactants having different congener profiles, they raised several questions that are worth investigating in the future. For instance, it remains to reveal how rhamnose is synthesized when HVGO is provided as the sole carbon source. Moreover, the mechanism of biosynthesis of monolipid congeners (those containing a single fatty acid moiety) deserves more systematic studies. It is also interesting to decipher how HVGO-triggered stress responses might affect the functioning of the quorum sensing cascades and what impact can this have on rhamnolipids biosynthesis.

## Figures and Tables

**Figure 1 molecules-26-04122-f001:**
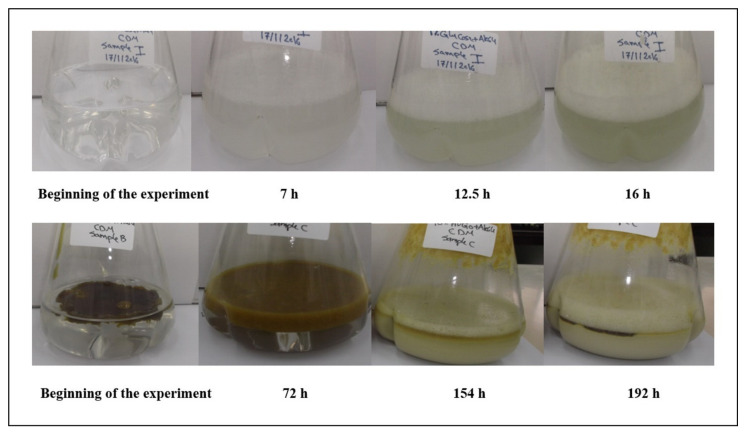
Growth of the strain AK6U on glucose and HVGO. The strain AK6U was cultured in chemically defined medium (CDM) containing either glucose (10 mM, upper panel) or HVGO (10% vol/vol, lower panel) as a sole carbon and energy source.

**Figure 2 molecules-26-04122-f002:**
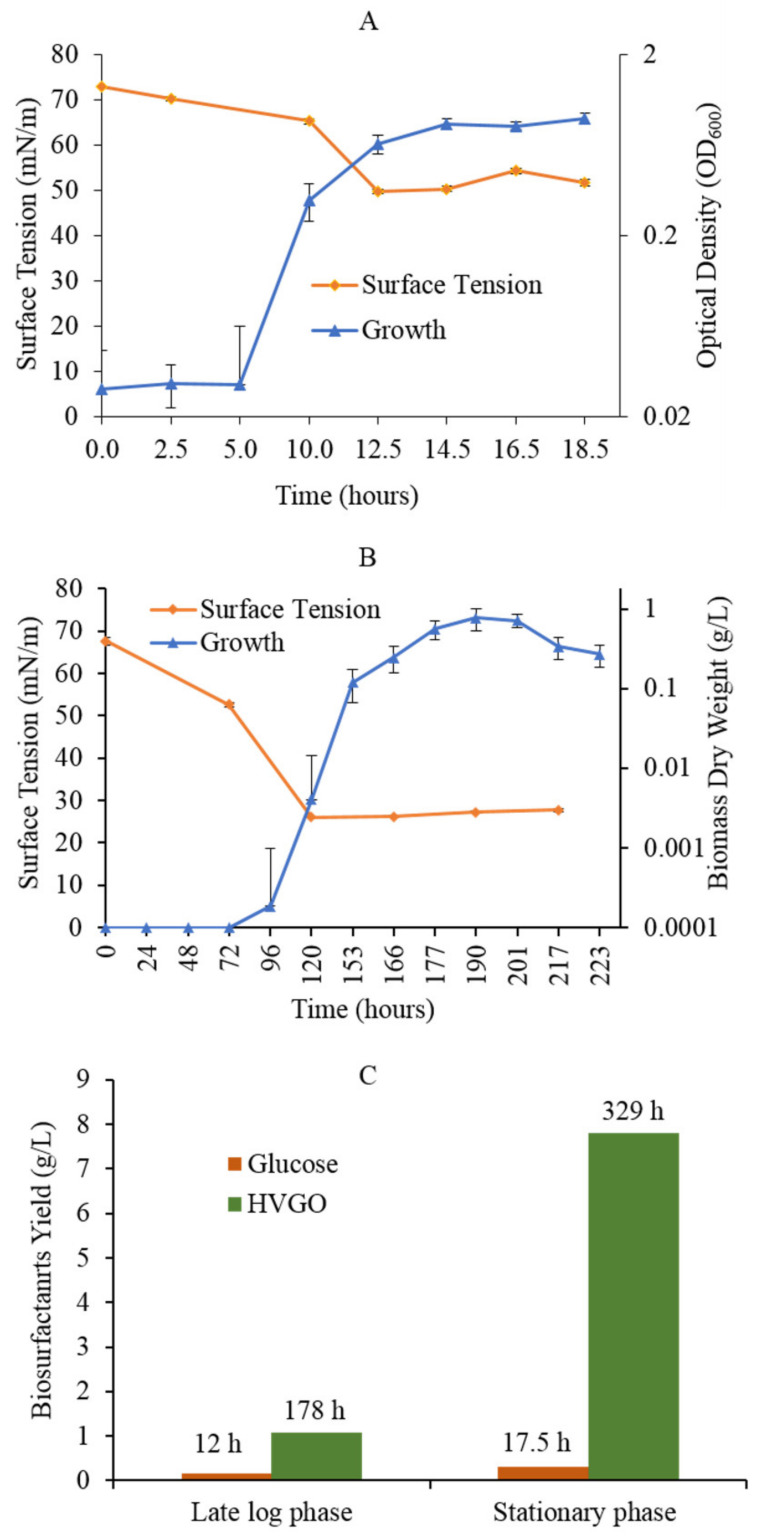
Growth profile and biosurfactants production kinetics of the glucose (**A**) and HVGO (**B**) cultures of the AK6U strain, and biosurfactants yield (**C**). Growth is plotted on a logarithmic scale. Time points at which biosurfactants were extracted from the cultures are indicated above the columns in panel C. Error bars represent standard error (number of biological replicates = 3).

**Figure 3 molecules-26-04122-f003:**
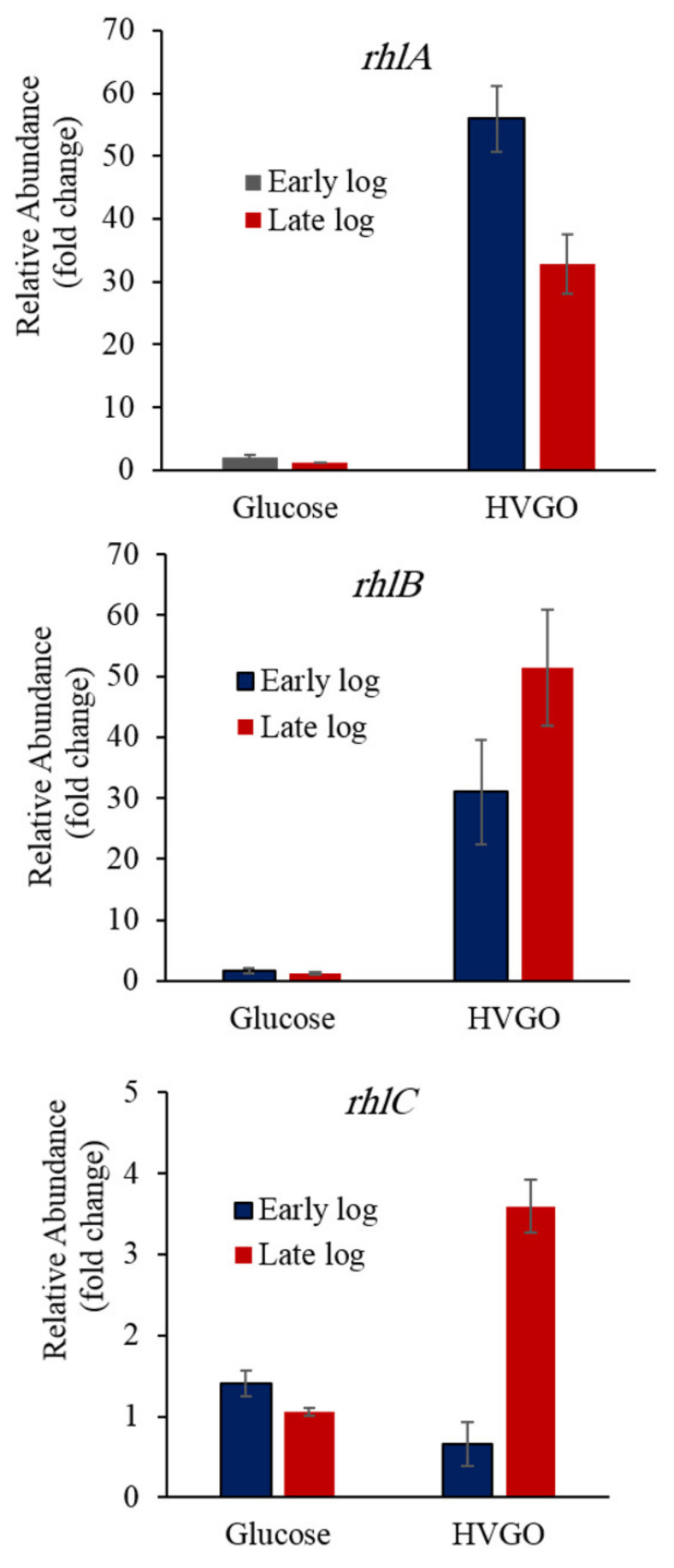
Relative expression of the *rhlABC* genes in the glucose and HVGO cultures of strain AK6U during the early and late log phases of growth. Error bars represent standard error (number of replicates = 2–6).

**Figure 4 molecules-26-04122-f004:**
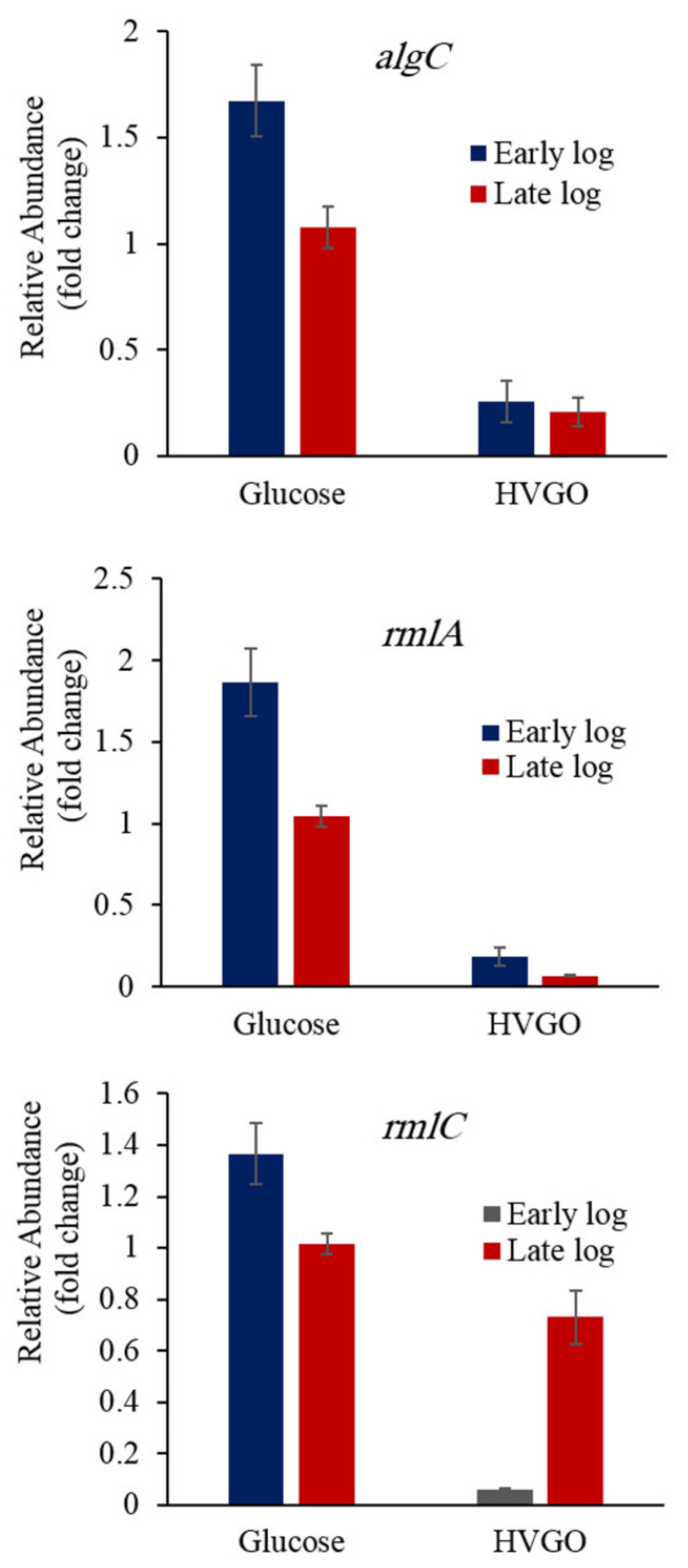
Relative expression of rhamnose biosynthesis genes in the glucose and HVGO cultures of strain AK6U during the early and late log phases of growth. Error bars represent standard error (number of replicates = 2–6).

**Figure 5 molecules-26-04122-f005:**
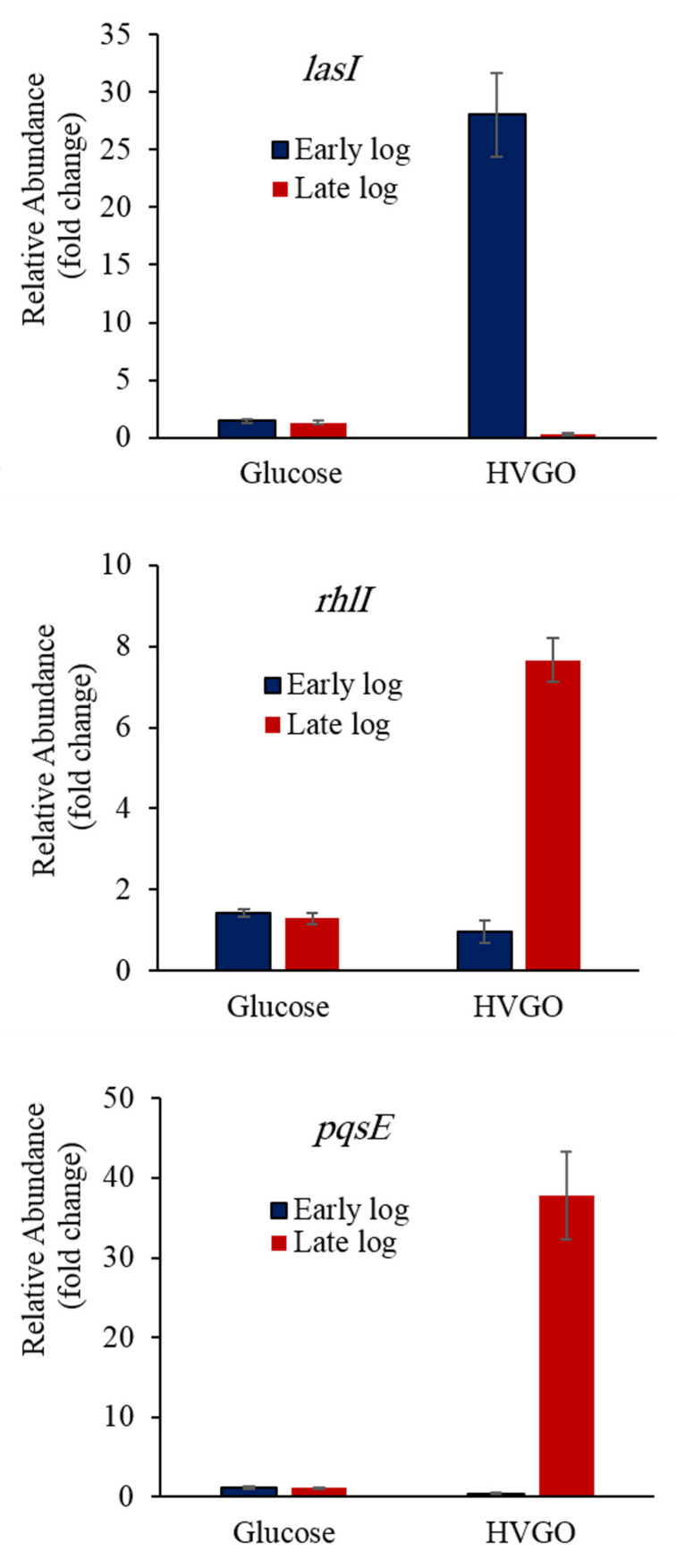
Relative expression of the quorum sensing genes in the glucose and HVGO cultures of strain AK6U during the early and late log phases of growth. Error bars represent standard error (number of replicates = 2–6).

**Table 1 molecules-26-04122-t001:** Relative abundance (%) of HAAs produced in glucose and HVGO cultures of the AK6U strain.

HAAs		Pseudo-Molecular Ion	Cultures ^#^		
			Glucose(S)	HVGO(L)	HVGO(S)
1	C_8_–C_8_	301	76.01		42.98
2	C_8_–C_10_	329	14.11		8.11
3	C_10_–C_8_	329	9.55		6.06
4	C_8_–C_12_	357			1.57
5	C_12_–C_8_	357			1.5
6	C_10_–C_10_	357			0.79
7	C_10_–C_12_	385			1.86
8	C_12_–C_10_	385			1.73
9	C_12_–C_12_	413			1.39
10	C_8_–C_12:1_	355			9.71
11	C_12:1_–C_8_	355			9.4
12	C_10_–C_12:1_	383			7.45
13	C_12:1_–C_10_	383			7.23
14	C_12_–C_12:1_	411	0.35		0.29
15	C_12:1_–C_12_	411			
	Total	15	4		14

^#^ L: late log phase, S: stationary phase.

**Table 2 molecules-26-04122-t002:** Relative abundance (%) of rhamnolipid congeners produced in glucose and HVGO cultures of the AK6U strain.

Rhamnolipid Congener		Pseudo-Molecular Ion	Cultures ^#^		
			Glucose(S)	HVGO(L)	HVGO(S)
1	Rha–C_8_	305	0.58		0.43
2	Rha–C_10_	333		0.08	
3	Rha–C_12_	361			
4	Rha–C_8_–C_8_	447	0.26		
5	Rha–Rha–C_8_	451	0.93		0.03
6	Rha–C_8_–C_10_	475	0.26	2.53	0.19
7	Rha–C_10_–C_8_	475		2.42	0.17
8	Rha–Rha–C_10_	479		0.76	3.48
9	Rha–C_10_–C_10_	503	2.54	14.33	3.43
10	Rha–Rha–C_12:1_	505	1.14	0.59	
11	Rha–Rha–C_12_	507	1.18		
12	Rha–C_10_–C_12:1_	529	2.35	1.11	0.01
13	Rha–C_12:1_–C_10_	529	0.74	1.01	
14	Rha–C_12_–C_10_	531		0.97	
15	Rha–C_10_–C_12_	531		0.91	
16	Rha–Rha–C_8_–C_8_	593	86.44		1.36
17	Rha–Rha–C_8_–C_10_	621	2.42	2.5	1.91
18	Rha–Rha–C_10_–C_8_	621	0.13	2.32	1.84
19	Rha–Rha–C_8_–C_12:1_	647			
20	Rha–Rha–C_12:1_–C_8_	647			
21	Rha–Rha–C_10_–C_10_	649	0.58	50	69.29
22	Rha–Rha–C_10_–C_12:1_	675		4.09	3.31
23	Rha–Rha–C_12:1_–C_10_	675		3.97	3.27
24	Rha–Rha–C_12_–C_10_	677	0.09	6.27	5.76
25	Rha–Rha–C_10_–C_12_	677		6.21	5.59
26	Rha–Rha–C_12:1_–C_12_	703	0.44		0.01
27	Rha–Rha–C_10_–C_14:1_	703			
28	Rha–Rha–C_12_–C_12_	705			
	Total	28	15	17	16

^#^ L: late log phase, S: stationary phase.

**Table 3 molecules-26-04122-t003:** Primers used in this study.

Gene	Locus	Primers	Sequence (5′-3′)	Amplicon Size (bp)	Source
*rhlI*	PA3476	RhlI-F RhlI-R	CTCTCTGAATCGCTGGAAGG GATGGTCGAACTGGTCGAAT	145	This study
*pqsE*	PA1000	PqsE-F PqsE-R	GACATGGAGGCTTACCTGGA CTCAGTTCGTCGAGGGATTC	197	This study
*lasI*	PA1432	LasI-F LasI-R	GGCTGGGACGTTAGTGTCAT GGCACGGATCATCATCTTCT	372	This study
*rmlA*	PA5163	RmlA-F RmlA-R	CCATCAGCCTGGAAGAGAAG GCTTGAGGTATTGGCCGTAG	390	This study
*rmlC*	PA5164	RmlC-F RmlC-R	CTTCGTCCAGGACAACCATT CACATCTGGCGCTTGTTCT	201	This study
*algC*	PA5322	AlgC-F AlgC-R	CCTACCCCGGTGCTGTACTA AGTCGACCACCACCTTCATC	280	This study
*rhlA*	PA3479	RhlA-F RhlA-R	TGGACTCCAGGTCGAGGAAA GAAAGCCAGCAACCATCAGC	263	[21]
*rhlB*		Kpd1 Kpd2	GCCCACGACCAGTTCGAC CATCCCCCTCCCTATGAC	226	[50]
*rhlC*	PA1130	RhlC-F2 RhlC-R2	GTCGAGTCCCTGGTTGAAGG CGTGCTGGTGGTACTGTTCA	211	[21]
16S rRNA		16S-F 16S-R	CACCGGCAGTCTCCTTAGAG AAGCAACGCGAAGAACCTTA	203	[21]

## Data Availability

The data presented in this study are contained in the manuscript and the Appendix A.

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
