# Peer review of "Heavy Vacuum Gas Oil Upregulates the Rhamnosyltransferases and Quorum Sensing Cascades of Rhamnolipids Biosynthesis in Pseudomonas sp. AK6U"

_molecules, 2021, doi:10.3390/molecules26144122_

Round 1

Reviewer 1 Report

The present study by Sarah Alkhalaf and collaborators is essentially the logical continuation of that published by Ismail et al. in 2017.  The authors explored in this study how heavy vacuum gas oil (HVGO), when used as a sole carbon source, affects Pseudomonas sp. AK6U growth, the mRNA levels of genes involved in rhamnolipid production and quorum sensing, as well as the composition of the rhamnolipids produced.  Knowledge derived from this study could lead to a better understanding of biological surfactant production from industrial oils, with a potential variety of applications.

From a biological standpoint, this is a challenging subject as growing in the presence of a complex mixture of (usually toxic) hydrocarbons like HVGO as a sole carbon source requires unique physiological tolerances, as well as specialised genetic and metabolic capacities.  These are physicochemical conditions to which Pseudomonads are particularly well-adapted, however, with mechanisms that are still obscure, notably at the genetic and metabolic levels.

The manuscript is very well written and structured, easy to read and follow, with sound methodologies and mostly clear results that support the conclusions made and raises some interesting questions for further research.

I suggest that the authors consider the following points in order to further improve their manuscript:

Lines 112-114, growth rates (µ) and doubling (generation) times (td): the values given have the relation µ = 1/td. However, 1/td corresponds to the doubling (generation) rate (v).  The correct relation between growth rate and generation time is µ = ln(2)v, or therefore µ = ln(2)/td.  Whichever parameter was determined first, µ or td, the other was not correctly calculated.

Figure 1: as bacterial growths, measured as OD600 or as biomass, are usually exponential, these should be plotted on semi-logarithmic scales.  For consistency and better clarity, the corresponding axes (and symbol legends) of growth and surface tension would benefit from being put on the same sides between the panels.

Lines 169-170: there is no data for Rha–Rha–C10–C12 in glucose in Table 3... It seems that this should have been Rha–Rha–C10–C10, as stated in line 332.  How significant could be the difference between Rha–Rha–C10–C10 (0.58%) and Rha–Rha–C12:1–C12 (0.44%), which looks closely similar, to state that the latter is not as abundant as the former?  It is unfortunate that these measures don't have some assessment of their uncertainty in order to make the claims made more robust.

Line 194: genomic instead of gnomic

Figure 5: it would be elegant to arrange the panels to reflect the classic hierarchy of quorum sensing depicted in Fig. S2, i.e. in the order: lasI, rhlI, pqsE.

Lines 373-379: these concepts are from microbiology textbooks, quite generic, and not really applicable to Pseudomonads, which are not known to normally produce starch or accumulate polysaccharide granules.  Although the authors pose an interesting question as to whether the glucose may have come from some reserve polymer, probably glycogen, this part would benefit from considering and citing the recently published paper by Woodcock et al. 2021, https://doi.org/10.1371/journal.pgen.1009524), particularly given that growing the cells in presence of 10% of HVGO potentially puts the cells under conditions that will induce physiological stress responses.

Lines 528-530: Why were two of the primers (for rhlAC, but what about the six others?) for RT-qPCR designed on the genomic sequence of strain PAO1 instead of that of AK6U, GenBank accession number CP025229?  If the sequences where the rhlAC primers anneal are rigorously identical between strains PAO1 and AK6U the latter would be preferable for the consistency of the manuscript, even if chronologically their design may not have happened that way.  If the sequences are not identical, this should be addressed and discussed. The sequences of these primers don't appear in Table 1, unless these are the already published RhlA-F and RhlC-R2, in which case that is not clear.

The table numbering is not consistent with a format in which the Materials and Methods section appears last (i.e. Table 1 is presently the last of the three).

Line 531: citing volatile website links instead of available publications is not a good idea.  At the end of the presently referred page ('Citing Primer3' at http://frodo.wi.mit.edu/primer3) the authors request that use of their software be citing their publications.  Citing Untergasser et al. 2012 should probably be enough.

Figure S2: for completeness, the legend should state as the two other quorum sensing systems explained, that the two signalling 2-alkyl-quinolones molecules are produced by the PqsABCD proteins (HHQ only, PQS requiring, in addition, the protein encoded by pqsH, which is located elsewhere in the chromosome) and then bind to their cognate transcriptional regulator, MvfR (PqsR).

Discussion: a theoretical aspect of this research that the authors may want to consider in their discussion is that HVGO being a highly hydrophobic mixture will act as a solvent and will therefore constitute a phase in which N-acyl-homoserine lactones and 2-alkyl-quinolones will partition from the aqueous growing medium, altering the dynamics of quorum sensing significantly compared to culture conditions without this signalling molecule sink, with the hydrophobicity of 2-alkyl-quinolones >> 3-oxo-C12-HSL >> C4-HSL.  However, this partitioning and loss of free quorum sensing signalling molecules in the growth phase should become mitigated over time by the emulsifying characteristics of rhamnolipids.  As far as I know, this has not yet been considered/studied/published but it could perhaps explain some of the peculiarities observed with the timing of expression of the studied quorum sensing genes.  It is also not excluded that the presence of certain molecules in HVGO (e.g. quinoline, tetrahydronaphthalene…) at high concentrations could interfere with the 'normal' metabolism and signalling processes observed in the glucose medium.

Author Response

I suggest that the authors consider the following points in order to further improve their manuscript:

Lines 112-114, growth rates (µ) and doubling (generation) times (td): the values given have the relation µ = 1/td. However, 1/td corresponds to the doubling (generation) rate (v).  The correct relation between growth rate and generation time is µ = ln(2)v, or therefore µ = ln(2)/td.  Whichever parameter was determined first, µ or td, the other was not correctly calculated.

Response: For the calculation of the specific growth rate, we used the following equation

µ= log (Xt)- log (X0)/0.301t

We have now removed the generation time from the manuscript to avoid confusion (lines 112-114).

Figure 1: as bacterial growths, measured as OD600 or as biomass, are usually exponential, these should be plotted on semi-logarithmic scales.  For consistency and better clarity, the corresponding axes (and symbol legends) of growth and surface tension would benefit from being put on the same sides between the panels.

Response: Growth was plotted on a logarithmic scale and the axes and symbols were aligned on the same side.

Lines 169-170: there is no data for Rha–Rha–C10–C12 in glucose in Table 3... It seems that this should have been Rha–Rha–C10–C10, as stated in line 332.  How significant could be the difference between Rha–Rha–C10–C10 (0.58%) and Rha–Rha–C12:1–C12 (0.44%), which looks closely similar, to state that the latter is not as abundant as the former?  It is unfortunate that these measures don't have some assessment of their uncertainty in order to make the claims made more robust.

Response: We apologize for this mistake. We corrected this statement (lines 166-167). Actually, there are no replicates for the rhamnolipid analyses. Therefore, it is not possible to assess how significant is the difference between the relative abundance 0.58% and 0.44%. In line 332, we are talking about the HVGO cultures.

Line 194: genomic instead of gnomic

Response: Corrected (line 191).

Figure 5: it would be elegant to arrange the panels to reflect the classic hierarchy of quorum sensing depicted in Fig. S2, i.e. in the order: lasI, rhlI, pqsE.

Response: The panels were rearranged

Lines 373-379: these concepts are from microbiology textbooks, quite generic, and not really applicable to Pseudomonads, which are not known to normally produce starch or accumulate polysaccharide granules.  Although the authors pose an interesting question as to whether the glucose may have come from some reserve polymer, probably glycogen, this part would benefit from considering and citing the recently published paper by Woodcock et al. 2021, https://doi.org/10.1371/journal.pgen.1009524), particularly given that growing the cells in presence of 10% of HVGO potentially puts the cells under conditions that will induce physiological stress responses.

Response: In this part, we tried to speculate on the potential sources of glucose/rhamnose. It is true that glycogen production is not well known in pseudomonads, but was reported for some species. We modified this part and cited the reference suggested by the reviewer in addition to other references (lines 376-380, 385-390).

Lines 528-530: Why were two of the primers (for rhlAC, but what about the six others?) for RT-qPCR designed on the genomic sequence of strain PAO1 instead of that of AK6U, GenBank accession number CP025229?  If the sequences where the rhlAC primers anneal are rigorously identical between strains PAO1 and AK6U the latter would be preferable for the consistency of the manuscript, even if chronologically their design may not have happened that way.  If the sequences are not identical, this should be addressed and discussed. The sequences of these primers don't appear in Table 1, unless these are the already published RhlA-F and RhlC-R2, in which case that is not clear.

Response: We apologize for the confusion. We have now clarified this issue in the text (lines 536-540) and inserted locus tags to Table 1. The genome sequence of the AK6U strain is not available. Therefore, we derived all sequences of the qPCR primers from the corresponding genes in the closely related rhamnolipid-producing PAO1 strain. All primer sequences are listed in Table 1. Some sequences were designed during this study as indicated and other primers were retrieved from the literature as indicated in Table 1.

The table numbering is not consistent with a format in which the Materials and Methods section appears last (i.e. Table 1 is presently the last of the three).

Response: We corrected and rearranged the table numbers

Line 531: citing volatile website links instead of available publications is not a good idea.  At the end of the presently referred page ('Citing Primer3' at http://frodo.wi.mit.edu/primer3) the authors request that use of their software be citing their publications.  Citing Untergasser et al. 2012 should probably be enough.

Response: Citation was added

Figure S2: for completeness, the legend should state as the two other quorum sensing systems explained, that the two signalling 2-alkyl-quinolones molecules are produced by the PqsABCD proteins (HHQ only, PQS requiring, in addition, the protein encoded by pqsH, which is located elsewhere in the chromosome) and then bind to their cognate transcriptional regulator, MvfR (PqsR).

Response: The figure legend was amended.

Discussion: a theoretical aspect of this research that the authors may want to consider in their discussion is that HVGO being a highly hydrophobic mixture will act as a solvent and will therefore constitute a phase in which N-acyl-homoserine lactones and 2-alkyl-quinolones will partition from the aqueous growing medium, altering the dynamics of quorum sensing significantly compared to culture conditions without this signalling molecule sink, with the hydrophobicity of 2-alkyl-quinolones >> 3-oxo-C12-HSL >> C4-HSL.  However, this partitioning and loss of free quorum sensing signalling molecules in the growth phase should become mitigated over time by the emulsifying characteristics of rhamnolipids.  As far as I know, this has not yet been considered/studied/published but it could perhaps explain some of the peculiarities observed with the timing of expression of the studied quorum sensing genes.  It is also not excluded that the presence of certain molecules in HVGO (e.g. quinoline, tetrahydronaphthalene…) at high concentrations could interfere with the 'normal' metabolism and signalling processes observed in the glucose medium.

Response: Thanks for drawing our attention to this point which is totally valid and for sure merits future investigation. We have now added some text to the discussion to highlight this issue (lines 435-445). There is no doubt that HVGO triggers various stress responses and alters the physiology and metabolism of the cell due to toxicity and hydrophobicity which should be the subject of future systems biology studies such as proteomics and transcriptomics.

Reviewer 2 Report

The authors have published an article that showed AK6U can utilize HVGO as the sole carbon and sulfur source to produce rhamnolipids. Here, the authors only compared the difference of the produced RLs and the transcriptional profiles when using glucose and HVGO. The difference of the produced RLs and the transcriptional profiles of the genes rhlABC and rhlA was  able to be predicted and the results displayed less significance.  I don't think that the novlty is enough. 

Author Response

Response: Even though the results were predictable, which is not necessarily correct, prediction is just a hypothesis that needs experimental work to be validated or refuted. Scientific research does not depend only on prediction. Our study is the first that reports detailed analysis of rhamnolipids and expression of the related genes in the presence of a challenging and interesting substrate like HVGO. Therefore, we are quite confident that the outcome extends our understanding of the field and merits publication. Moreover, our study raised some more interesting questions that will fuel further studies.

Reviewer 3 Report

  1. Figure 2: please change the axes so that the surface tension is always on the same side in each graph (e.g. on the right vertical axis). It would be much easier to compare the graphs.
  2. Figure 2: why the optical density was not measured for more than 18.5 h? Especially that the dry weight was determined for 223 h? How was the late log phase determined for glucose? Only on the basis of dry matter? Graph C - please add the cultivation time (hours) in which the biosurfactants yield was determined. Why the same parameters were not determined for both cultures (OD and biosurfactants yield for both glucose and HVGO)?
  3. Figure 2: why there are no error bars in graph C? Was biosurfactants yield not determined in duplicates/triplicates? It is not reliable if there were no replicates and only one culture with glucose and one with HVGO was performed...
  4. Table 2 and 3: why the relative abundance (%) of late log phase of glucose broth was not determined? There might be some changes...
  5. Figure 4 should be placed in subsection “2.6. Downregulation of the rhamnose biosynthesis genes in the HVGO culture ”as it is described there.
  6. Where was the HVGO obtained from?
  7. Why was the dry weight of cells determined on the basis of 25 ml of culture? What was the determinant factor used here? Was it CFU?
  8. Although significant changes of surface tension and biosurfactants yield are visible, the cultures with HVGO are much longer than with glucose. That raises the question: how is it economically viable? For the industrial production of biosurfactant? Did the Authors plan any changes in the cultivation to shorten the time needed to obtain the biosurfactant?
  9. It is always worth referring to the other biosurfactants, not only those obtained from Pseudomonas (in the discussion). A table comparing the surface tension of other bacterial biosurfactants with this studied in the manuscript would be useful.
  10. Please place the Conclusions section after the Materials and Methods section.

Additionally, please take care of the following points:

  1. Line 99: "gaps" instead of "gabs"
  2. Line 194: "genomic" instead of "gnomic"
  3. Line 376: use numerical references instead of "(Madigan et al., 2009)"
  4. Line 456: Please write “30°C” instead of “30 ° C” - that applies to the whole manuscript. Additionally, pay attention to the unnecessary spaces in the text.
  5. Line 476: "via the Wilhelmy plate method" - no reference, please complete.

Author Response

  1. Figure 2: please change the axes so that the surface tension is always on the same side in each graph (e.g. on the right vertical axis). It would be much easier to compare the graphs.

Response: The axes were rearranged

2. Figure 2: why the optical density was not measured for more than 18.5 h? Especially that the dry weight was determined for 223 h? How was the late log phase determined for glucose? Only on the basis of dry matter? Graph C - please add the cultivation time (hours) in which the biosurfactants yield was determined. Why the same parameters were not determined for both cultures (OD and biosurfactants yield for both glucose and HVGO)?

Response: The glucose culture entered the stationary phase after and there was no need to further measure the OD. The late log phase for the glucose culture was determined from the growth curve to be 12 hours. The presence of HVGO and the strong emulsification interferes with OD measurement. Therefore, in the HVGO culture we followed the growth by quantifying the biomass dry weight and we have even continued to measure beyond 223 (not shown) to ensure the culture has reached a plateau. We added the cultivation time to Graph C. 

3. Figure 2: why there are no error bars in graph C? Was biosurfactants yield not determined in duplicates/triplicates? It is not reliable if there were no replicates and only one culture with glucose and one with HVGO was performed...

Response: The measurement was done once, that is why there are no error bars. Although we agree with the reviewer that replicate measurements would have been better, the results are still reliable because the difference between both cultures is large. Furthermore, the results are in line with the gene expression and rhamnolipid analysis.

4. Table 2 and 3: why the relative abundance (%) of late log phase of glucose broth was not determined? There might be some changes...

Response: There is no notable difference in biosurfactants yield between the two phases. Therefore, we hypothesized that there is no remarkable differences at the level of the congener profile and we focused only on the stationary phase culture. 

5. Figure 4 should be placed in subsection “2.6. Downregulation of the rhamnose biosynthesis genes in the HVGO culture ”as it is described there.

Response: The location of the figure was corrected

6. Where was the HVGO obtained from?

Response: From the oil refinery of Bahrain

7. Why was the dry weight of cells determined on the basis of 25 ml of culture? What was the determinant factor used here? Was it CFU?

Response: There is no specific reason. We could have measured it in more or less volume.

8. Although significant changes of surface tension and biosurfactants yield are visible, the cultures with HVGO are much longer than with glucose. That raises the question: how is it economically viable? For the industrial production of biosurfactant? Did the Authors plan any changes in the cultivation to shorten the time needed to obtain the biosurfactant?

Response: Thanks for raising this point since the ultimate goal of this kind of research is to develop an economically viable bioprocess for biosurfactants production. However, this needs extensive work to optimize the biosurfactants production conditions which is planned in our laboratory. HVGO is a challenging and complex substrate that needs handling and analytical tools different from those needed for glucose. Eventually, there should be a kind of trade off considering the pros and cons of the process.

9. It is always worth referring to the other biosurfactants, not only those obtained from Pseudomonas (in the discussion). A table comparing the surface tension of other bacterial biosurfactants with this studied in the manuscript would be useful.

Response: Comparison of the surface tension values attained by different biosurfactants in different studies can be sometimes inconclusive because of differences in the type of the producing microbe, the culture conditions, extraction protocol, and downstream purification steps. We added some text to the discussion referring to surface tension values of rhamnolipids reported in the literature (lines 286-289).

10. Please place the Conclusions section after the Materials and Methods section.

 Response: Done

Additionally, please take care of the following points:

  1. Line 99: "gaps" instead of "gabs"
  2. Line 194: "genomic" instead of "gnomic"
  3. Line 376: use numerical references instead of "(Madigan et al., 2009)"
  4. Line 456: Please write “30°C” instead of “30 ° C” - that applies to the whole manuscript. Additionally, pay attention to the unnecessary spaces in the text.
  5. Line 476: "via the Wilhelmy plate method" - no reference, please complete.

Response: All corrections were done

Round 2

Reviewer 2 Report

The authors have revised the manuscript well according to the reviewers’ comments and suggestions. I recommend acceptance.

Reviewer 3 Report

The manuscript has been improved by addressing the Reviewer's comments. This paper is now acceptable for publication in  Molecules.